# Investigating the Influence of Diverse Functionalized Carbon Nanotubes as Conductive Fibers on Paper-Based Sulfur Cathodes in Lithium–Sulfur Batteries

**DOI:** 10.3390/nano14060484

**Published:** 2024-03-07

**Authors:** Xuan Ren, Haiwei Wu, Ya Xiao, Haoteng Wu, Huan Wang, Haiwen Li, Yuchen Guo, Peng Xu, Baohong Yang, Chuanyin Xiong

**Affiliations:** Shaanxi Provincial Key Laboratory of Papermaking Technology and Specialty Paper Development, College of Bioresources Chemical and Materials Engineering, Shaanxi University of Science & Technology, Xi’an 710021, China; 210111018@sust.edu.cn (X.R.); xiaoya1@stu.scu.edu.cn (Y.X.); 220112099@sust.edu.cn (H.W.); 220112146@sust.edu.cn (H.W.); 220112201@sust.edu.cn (H.L.); 220112153@sust.edu.cn (Y.G.); 220112096@sust.edu.cn (P.X.); xiongchuanyin@sust.edu.cn (C.X.)

**Keywords:** lithium–sulfur batteries, paper-based electrodes, carbon nanotubes

## Abstract

Lithium–sulfur (Li–S) batteries are expected to be one of the next generations of high-energy-density battery systems due to their high theoretical energy density of 2600 Wh kg^−1^. Embracing the trends toward flexibility, lightweight design, and cost-effectiveness, paper-based electrodes offer a promising alternative to traditional coated cathodes in Li–S batteries. Within paper-based electrodes, conductive fibers such as carbon nanotubes (CNTs) play a crucial role. They help to form a three-dimensional network within the paper matrix to ensure structural integrity over extended cycling while mitigating the shuttle effect by confining sulfur within the cathode. Herein, we explore how variously functionalized CNTs, serving as conductive fibers, impact the physical and electrochemical characteristics of paper-based sulfur cathodes in Li–S batteries. Specifically, graphitized hydroxylated carbon nanotubes (G-CNTs) exhibit remarkable capacity at low currents owing to their excellent conductivity and interaction with lithium polysulfide (LiPS), achieving the highest initial specific capacity of 1033 mAh g^−1^ at 0.25 C (1.1 mA cm^−2^). Aminated multi-walled carbon nanotubes (NH_2_-CNTs) demonstrate an enhanced affinity for LiPS due to the -NH_2_ groups. However, the uneven distribution of these fibers may induce electrode surface passivation during charge–discharge cycles. Notably, hydroxylated multi-walled carbon nanotubes (OH-CNTs) can establish a uniform and stable 3D network with plant fibers, showcasing superior mechanical properties and helping to mitigate Li_2_S agglomeration while preserving the electrode porosity. The paper-based electrode integrated with OH-CNTs even retains a specific capacity of approximately 800 mAh g^−1^ at about 1.25 C (5 mA cm^−2^), demonstrating good sulfur utilization and rate capacity compared to other CNT variants.

## 1. Introduction

In recent years, with the rapid development of wearable electronics, there has been an increasing demand for flexible and lightweight energy storage devices [1,2,3,4,5,6,7]. These electronic devices exhibit unique advantages such as mechanical stability, high flexibility, and excellent portability, possessing broad application prospects in areas such as electronic skin, flexible displays, wearable devices, etc. It is foreseeable that flexible electronic technologies will trigger a new round of the electronic technology revolution and influence future lifestyles [8,9,10,11]. However, these characteristics of wearable devices require batteries to be flexible, thin, and lightweight, which will inevitably sacrifice energy density to some extent. Traditional lithium-ion batteries (LIBs) are now commonly used in portable electronic products, but their energy density is nearing the theoretical limit of around 420 Wh kg^−1^, and the development of battery systems with higher gravimetric energy density is imminent [12]. Lithium–sulfur batteries (LSBs), which store and release energy through a reversible electrochemical reaction between lithium and sulfur, have an energy density of 2600 Wh kg^−1^ and high theoretical specific capacity of 1675 mAh g^−1^. This potential makes them highly promising candidates to potentially surpass traditional lithium-ion batteries (LIBs) [13,14]. However, Li–S batteries still suffer from many problems that restrict their commercialization and especially limit the utilization of flexible energy storage devices. Firstly, the insulating nature of sulfur and lithium sulfide hinders electron transport, resulting in limited reaction kinetics and reduced active material utilization, thereby impacting the real energy density of the battery [15,16,17]. Secondly, the reaction intermediates, the medium-chain and long-chain lithium polysulfides (LiPS) soluble in the electrolyte, induce the shuttle effect, which is the primary cause of capacity degradation and low Coulombic efficiency [15,18,19]. Moreover, due to the density disparity between sulfur and lithium sulfide, the sulfur cathodes will undergo dramatic volume changes during cycling, resulting in electrode structure damage or even battery failure [15,20,21]. In addition to these common issues in conventional Li–S batteries, the development of flexible Li–S batteries is also constrained by the flexibility of the electrodes [22]. Electrodes prepared using the conventional coating process may not meet the flexibility requirements of such batteries, potentially leading to delamination of the active material layer and electrode fracture upon bending [15,23].

To address these challenges, substantial efforts have been devoted to enhancing the design of flexible sulfur cathodes [22]. Various approaches have been explored, including the development of multifunctional binders tailored for flexible electrodes [15,24], the integration of composites with substrates to create flexible electrodes [25], the infiltration of sulfur into pre-fabricated flexible frameworks to realize flexible sulfur cathodes [26,27,28], etc. Among the abundant strategies for preparing flexible sulfur cathodes, paper-based electrodes are widely favored due to its advantages, such as its low cost, lightweight nature, and eco-friendliness. Contemporary studies frequently integrate conductive materials such as carbon nanotubes, graphene, and carbon fibers into paper-like film structures [29,30,31]. Such self-supporting paper-based electrodes combine superior flexibility with excellent electrochemical performance. However, for the high capacity of Li–S batteries, they must also exhibit strong LiPS affinity to mitigate the shuttle effect and provide sufficient space for accommodating volume expansion [32].

Considering this, our group has tried variable paper-based electrodes featuring a 3D fiber network structure and good LiPS affinity by means of the papermaking process [32,33,34]. This 3D fiber network leverages the mesh structure formed by interlaced and lapped plant fibers to constitute the main skeleton of the electrode for mechanical support. The introduction of nanocellulose and bacterial cellulose with smaller dimensions, a larger specific surface area, and abundant surface -OH groups facilitate fiber entanglement and the formation of numerous hydrogen bonds, thereby significantly enhancing the mechanical properties of the paper-based electrode. This 3D network structure is also better suited to encapsulating micro-powder particles, reducing the shedding of active substances when bending or folding. In addition, the mechanically stable skeleton and porous configuration of the paper-based electrodes can adapt to the volume expansion and LiPS adsorption during the charging and discharging process [35]. Notably, CNTs play a crucial role as conductive agents, significantly contributing to the mechanical and electrochemical characteristics of paper-based sulfur cathodes. Therefore, we selected CNTs with diverse functionalizations, possessing a length of more than 50 μm and a tube diameter of 8–15 nm, to serve as the conductive fibers. Moreover, an excellent length–diameter ratio of CNTs facilitates their integration into the 3D fiber network structure, intertwining with plant fibers to improve the strength of the paper-based electrode. Meanwhile, the distinct functional groups on the surface of carbon nanotubes affect the mechanical strength of the paper-based electrode, the adsorption and conversion of LiPS, and the reaction kinetics of Li–S batteries. 

Herein, we find that the electrochemical performance of graphitized hydroxylated carbon nanotubes (G-CNT) is good at low discharge/charge currents due to the good conductivity and a slight surface modification of -OH, but is not suitable for high-current conditions. An aminated multi-walled carbon nanotube (NH_2_-CNT) has polar functional groups -NH_2_ that can interact with LiPS. However, it has a negative effect on battery performance due to the uneven distribution of NH_2_-CNTs when preparing the electrodes, which leads to passivation on the electrode surface during long charge–discharge cycles. Besides, because of the large number of -OH groups, hydroxylated multi-walled carbon nanotubes (OH-CNTs) are well-dispersed during the preparation of paper-based electrodes, resulting in the formation of a uniform and stable three-dimensional network structure with plant fibers, which exhibits superior mechanical properties and prevents the agglomeration of Li_2_S. The Li–S cells containing OH-CNTs deliver the highest capacity of 802 mAh g^−1^ at 5 mA cm^−2^ with 1.99 mg cm^−2^ sulfur loading, which shows excellent electrochemical performance compared to other CNTs. In summary, this study explores the potential of improving the physical and electrochemical properties of flexible sulfur cathodes by investigating the diverse functionalized conductive fibers. The findings hold profound significance for the further application and development of flexible paper-based electrodes in Li–S batteries.

## 2. Experimental Sections

### 2.1. Materials

Hardwood fiber (HWF) was purchased from Kimberly-Clark (China) Co., Ltd., Shanghai, China and processed in the laboratory to a beating degree of 90°RS. Cellulose nanofilament (CNF, 1.0 wt%, Φ: 4~10 nm, L: 1~3 μm) and bacterial nanocellulose fiber (BNF, 0.8 wt%, Φ: 50~100 nm, L: ~20 μm) were purchased from Guilin Qihong Technology Co., Ltd., Guilin, China. Multi-walled carbon nanotubes (MWCNT, L: ~50 μm), hydroxylated multi-walled carbon nanotubes (OH-CNT, L: ~50 μm), aminated multi-walled carbon nanotubes (NH_2_-CNT, L: ~50 μm), nickel-plated carbon nanotubes (Ni-CNT, L: ~50 μm), and graphitized hydroxylated carbon nanotubes (G-CNT, L: ~50 μm) were purchased from Beijing Deke Daojin Science And Technology Co., Ltd., Beijing, China. Sulfur nanopowder (S_8_, Φ: 30 nm) was purchased from Beijing Deke Daojin Science And Technology Co., Ltd., Beijing, China. Sodium dodecylbenzene sulfonate (SDBS, C_12_H_25_SO_4_Na, 99.5%) was purchased from Tianjin Damao Chemical Reagent Factory, Tianjin, China. Polyethylene glycol (PEG-200) was purchased from Shanghai Aladdin Biochemical Technology Co., Ltd., Shanghai, China. Electrolyte (1M LiTFSI in DME:DOL = 1:1 Vol% with 2% LiNO_3_) was purchased from Suzhou Forsyth Technology Co., Ltd., Suzhou, China. A PP diaphragm (Celgard2500, 25 μm) was purchased from Boliton Diaphragm Products Co., Ltd., Shanghai, China. The above raw materials are industrial products.

### 2.2. Preparation of Flexible Paper-Based Sulfur Cathodes

A total of 25 mg of nano S, 2 mL of PEG-200, and 15 mg of SDBS was added into deionized water with a volume of 15mL and stirred continuously for about 5 h until the nano sulfur was well dispersed, and the suspension appeared to be white. Another 15 mL of deionized water was taken, and 2 mg of HWF, 90 mg of BNF with a content of 0.8%, and 15 mg of CNF with a content of 1% were added, stirring to be well dispersed, then 15 mg of CNT and 15 mg of SDBS were added, and the solution was ultrasonically pulverized for 3 min. The electrode slurry was obtained by mixing the ultrasonic solution with the initially dispersed sulfur solution. After vacuum filtration and drying under vacuum at 60 °C, −0.1 MPa for 5 h, the flexible paper-based sulfur cathodes were obtained.

In the preparation process, the CNTs added include MWCNT, Ni-CNT, OH-CNT, NH_2_-CNT, and G-CNT, resulting in different paper-based sulfur cathodes.

### 2.3. Preparation of Sulfur-Free Flexible Paper-Based Electrodes

A total of 25 mg of SDBS, 2 mg of HWF, 90 mg of BNF with a content of 0.8%, and 15 mg of CNF with a content of 1% were added into deionized water with a volume of 30 mL and stirred until the fibers were dispersed, then 15 mg of CNT and 25 mg of SDBS were added. After ultrasonically pulverizing the solution for 3 min, the electrode slurry was obtained by stirring the solution well. The resulting slurry was vacuum-pumped and filtered, and then dried under vacuum at 60 °C, −0.1 MPa for 5h to obtain sulfur-free flexible paper-based electrodes.

In the preparation process, the CNTs added were MWCNT, Ni-CNT, OH-CNT, NH_2_-CNT, and G-CNT, resulting in different sulfur-free paper-based electrodes.

### 2.4. Characterization and Measurements

The thickness was measured with an electronic spiral micrometer. The test of electrical conductivity was completed by a four-probe instrument (Four Probes Tech, Guangzhou, China, RTS-9). The tensile strength and elongation at break were tested by a high- and low-temperature servo tension machine (Gotech, Shanghai, China, AI-7000NGD). The morphologies of the electrodes were observed using a scanning electron microscope (SEM) (Beijing, China, JEOL S-4800). UV-Vis measurements of Li_2_S_6_ solution before and after the addition of MWCNT, Ni-CNT, OH-CNT, NH_2_-CNT, and G-CNT were carried out within the range of 380–800 nm using a Cary 5000 UV/Vis spectrophotometer. A CR2016 coin-type Li–S battery was fabricated by assembling a self-supported paper-based sulfur cathode, a lithium metal anode, a commercial polypropylene (PP) separator, and a 1M LiTFSI-DME/DOL electrolyte with 2 wt% LiNO_3_. Constant current charge–discharge tests and lithium sulfide deposition tests were conducted on the battery using a battery tester (Shenzhen, China, NEWARE CT-4008T). The cyclic voltammetry (CV) and electrochemical impedance spectroscopy (EIS) of the battery and symmetrical cells were measured using an electrochemical workstation (Shanghai, China, CHI760E). All electrochemical tests of the battery were performed at room temperature.

## 3. Results and Discussion

In this study, flexible paper-based sulfur cathodes were prepared using a conventional paper-making process by mixing HWFs, CNFs, and BNFs as the supporting skeleton and CNTs as the conductive fibers with nano sulfur. As shown in Figure 1 and Appendix A, HWFs form the foundation of the flexible skeleton, while the addition of BNFs and CNFs enhances the hydrogen bonding among the fibers, resulting in a robust 3D fiber network structure that significantly enhances the mechanical durability of the paper-based electrodes and improves their ability to encapsulate nano-sulfur particles. Additionally, CNTs not only function as a conductive agent but also possess an optimal aspect ratio, allowing them to interlace with the fibers, thereby further improving the mechanical resilience of the paper-based electrodes. We introduced different functionalized CNTs in paper-based sulfur cathodes to explore the possibility of surface-functionalized CNTs in improving the conductivity, mechanical strength and Li–S battery performance.

The scanning electron microscopy (SEM) images in Figure 2 illustrate the different surface morphology of CNT electrodes. It can be clearly seen that the components on the surface of OH-CNT electrodes and G-CNT electrodes are most uniformly distributed, and the carbon tubes and fine plant fibers are obviously in the form of a filamentary network, enveloping the stacked nano sulfur. In contrast, although the 3D fiber network structure can still be observed in the NH_2_-CNT electrode, a few NH_2_-CNT agglomerates also appear in some areas, which may be attributed to the fact that -OH groups are more hydrophilic than -NH_2_ groups, causing the hydroxyl group-modified OH-CNTs and G-CNTs to be more adequately dispersed in water during the electrode pulp preparation. Comparatively, the MWCNT and Ni-CNT electrodes do not possess hydrophilic functional groups and the conditions for the formation of a large number of hydrogen bonds, resulting in the severe agglomeration of CNTs on the surface of the electrodes and the incomplete fiber network. In particular, Ni-CNTs display a certain magnetic property, which is more likely to be clustered together in electrode pulp. When the distribution of the components is uneven and the conductive fiber cannot effectively bridge and combine with other fibers to form a network structure, all aspects of the performance of the paper-based electrode will be affected.

The conductivities of the CNT electrodes tested by the four-probe instrument are shown in Figure 3a. Influenced by a large number of agglomerations of Ni-CNT, the Ni-CNT electrode fails to form a good conductive network, hence exhibiting the worst conductivity. MWCNTs have a wider distribution in the cathode due to their largest apparent volume, which ensures an unobstructed conductive pathway inside the electrode. The G-CNTs have better conductivity because of the graphitized treatment, so the cathode containing G-CNTs also shows excellent conductivity. While the surface modifications -OH and -NH_2_ may reduce the conductivity of CNTs, the conductivity of the electrodes prepared by OH-CNT and NH_2_-CNT is unsatisfactory. Only the fast electron migration in the cathodes can ensure the fast conversion of sulfur and lithium sulfide, which in turn improves the utilization rate of the active material.

Meanwhile, the adsorption and transportation of LiPS on the surface of the cathode are also very important, as they will directly affect the kinetic process of the Li–S battery [36]. To test the difference in LiPS interactions, adsorption tests were carried out by adding 0.1 g of each CNTs into 5 mL vials containing 5 mM Li_2_S_6_-DME/DOL solution. As shown in Figure 3b, the Li_2_S_6_ solution samples with OH-CNT and NH_2_-CNT reveal the most obvious discoloration over time. To further investigate the adsorption effect on LiPS of CNTs and quantitatively analyze the color change, sample solutions after adsorption for seven days were taken for UV-visible absorption spectroscopy, and the results are shown in Figure 3c. The absorbance of the S_4_^6−^ peak drops to much lower intensities after the addition of the NH_2_-CNT and OH-CNT, indicating they have stronger interactions with LiPS than other CNTs. This is attributed to the -OH and -NH_2_, which are polar groups with good affinity for LiPS [37,38,39,40,41]. On the other hand, G-CNTs display a poor absorption of LiPS, which may be because the amount of -OH groups modified on the G-CNT surface is not as high as OH-CNTs. Sufficient porosity inside the electrode not only facilitates the penetration of the electrolyte which can provide a good ion transport channel, but also accommodates the volume expansion of sulfur during the charge–discharge process. The porosities (*P*) of the electrode in this study are calculated using Equation (1):*P* = *(V*_0_ − *V)*/*V*_0_ × 100%(1)
where *V*_0_ refers to the volume of the material in its natural state or apparent volume, and *V* means the absolute dense volume of the material. Since the true density of CNTs is approximate, *V* can be regarded as a constant value in this experiment, and the main factor affecting the porosity is the thickness of the electrode sheet related to *V*_0_. The thicknesses and calculated porosities of the paper-based electrodes prepared with different CNTs are shown in Figure 3d. It can be seen that all electrodes exhibit good porosity except for the electrode containing Ni-CNTs, which can be attributed to the wonderful 3D fiber network structure, reinforcing the encapsulation and binding of powder particles such as nano sulfur, avoiding tiny particles from filling up the internal pores of the electrode.

The 3D fiber networks not only ensure sufficient pore space inside the electrode but also the acceptable mechanical properties of the paper-based electrodes. Figure 3e shows the tensile strength curves of the paper-based electrodes. It can be seen that among all paper-based electrodes, the tensile strength of OH-CNT is the largest at 2.73 MPa, and the elongation at break point is 3.25%. The tensile strength and elongation of NH_2_-CNT are slightly inferior to that of OH-CNT, but still far superior to other CNTs. Although G-CNTs have a small amount of -OH groups on the surface, they are not sufficient to form a large number of hydrogen bonds to consolidate the connection of different fibers in the G-CNT cathode [42]. That is also the reason why the mechanical properties of the G-CNT cathode are better than those of unfunctionalized CNTs, but not as good as OH-CNT. Figure 3f demonstrates the images of the electrode fracture sites after stretching. The presence of coarse and fine fibers can be observed, which further proves that the 3D fiber network plays a key protective role when the electrode is subjected to stress.

Following these tests, we found that when different functionalized CNTs are utilized as conductive fibers in flexible sulfur electrodes, they each demonstrate distinct advantages in improving the conductivity, LiPS adsorption, porosity, and mechanical properties of the paper-based electrodes. To further investigate the effect of the conductive fibers on the kinetics of LiPS conversion, free-standing CNT papers without sulfur were prepared and implemented as current collectors for the LiPS- conversion reaction. Symmetric cells were assembled with two identical papers as electrodes in 40 µL 0.5 mol L^−1^ Li_2_S_6_. The electrochemical impedance spectra (EIS) in Figure 4a demonstrate different resistances of these samples. The EIS curves of MWCNT, Ni-CNT, and G-CNT all consist of a semicircle in the high-frequency region and a diagonal line in the low-frequency region, where the diameter of the semicircle correlates with the charge transfer resistance (R_ct_) of the cathodes, and the diagonal line in the low-frequency region corresponds to the diffusion of ions [32]. Among them, G-CNT has the smallest semicircle, indicating that it has a much lower R_ct_ for liquid LiPS conversion. However, for NH_2_-CNT and OH-CNT, a new semicircle appears in the high-frequency region, referring to the electrode/electrolyte interface film impedance (R_sf_), reflecting the reaction and diffusion processes of sulfides on the electrode surface [43,44]. R_ct_ in the mid-frequency region is related to the solid-state insulating products on the conductive substrate surface (consumption of sulfur and accumulation of lithium sulfide) and the concentration variation of the reactant LiPS in the electrolyte. In a symmetric cell, the concentration of Li_2_S_6_ in the electrolyte is relatively high, resulting in a larger R_ct_. However, in a Li–S cell, the active material particles on the positive electrode surface are not activated before discharge, leading to a lower LiPS concentration in the electrolyte and thus a smaller R_ct_. However, as the electrolyte contacts and decomposes on the lithium anode, forming a passivation layer on the lithium anode surface, R_sf_ will increase. This is also the reason for the differences in EIS testing between symmetric and full cells with OH-CNTs and NH_2_-CNTs as shown in Figure 4a,b. In the case of NH_2_-CNT full cells, there may be some self-discharge phenomenon, leading to a higher concentration of LiPS in the electrolyte and a corresponding increase in R_ct_.

Cyclic voltammetry (CV) tests of symmetrical cells were conducted in the voltage range −1.5~1.5V at 1 mV s^−1^ in Figure 4c. The high capacitive current response and small peak separation between the reduction and oxidation peaks both demonstrate fast and reversible conversion of LiPS [45]. Comparing the capacitive current responses, it is clear that the symmetric cell with G-CNT paper exhibits a much higher current response than other CNTs. In addition, the symmetric cell of OH-CNT paper significantly shows the narrowest peak separation, confirming that the OH-CNT can effectively enhance the kinetics of the lithiation/delithiation reactions for polysulfide conversion [45]. This can be attributed to the positive effect of -OH groups on the conversion of long-chain polysulfides, which further mitigates the shuttle effect [46]. Meanwhile, it is also noteworthy that the peaks of OH-CNT and NH_2_-CNT are split, which implies more rapid and sufficient LiPS conversion on the electrode surface [47]. CV tests of full cells were also conducted on Li–S batteries to further confirm the kinetic enhancement capabilities of each CNT. Figure 4d–h and Appendix A display the CV curves of Li–S batteries assembled with paper-based cathodes at various scan rates. As shown in Figure 4d, the reduction peak corresponds to the reduction of S_8_ to soluble LiPS and LiPS to solid Li_2_S_2_/Li_2_S, while the oxidation peak represents the oxidation of long-chain LiPS to Li_2_S_2_/Li_2_S and LiPS to S_8_. It can be observed that G-CNTs shows less polarization at all scan rates than other CNTs. Moreover, the potential difference ΔE between the reduction and oxidation peaks can be used to reflect the polarization at different scan rates, where a smaller ΔE corresponds to lower polarization and faster reaction kinetics. As depicted in Figure 4e–h and Appendix A, with increasing scan rates, although the peak separation increases, G-CNT, OH-CNT, and NH_2_-CNT batteries consistently exhibit lower polarization for the reduction and oxidation processes, indicating their suitable catalytic activities in effectively promoting the rapid oxidation–reduction kinetics of sulfur, LiPS, and Li_2_S. Additionally, the strong current response of the reduction and oxidation peaks of G-CNT is attributed to its enhanced catalytic activity, demonstrating that G-CNT can effectively facilitate the rapid reversible conversion between sulfur, LiPS, and Li_2_S_2_/Li_2_S in Li–S batteries.

On the other hand, combining the Randles–Sevcik Equation (2):*I_p_ =* (2.65 *×* 10^5^) *n*^1.5^ *S D_Li+_*^0.5^ Δ*C_Li_ v*^0.5^(2)
where *I_p_* refers to the peak current (mA), *n* represents the number of electrons transferred in redox processes, *S* means the geometric area of the anode (cm^2^), *D_Li+_* stands for the lithium-ion diffusion coefficient (cm^2^ s^−1^), *v* means scan rate (V s^−1^), and Δ*C_Li_* represents the change of Li^+^ concentration during electrochemical reaction (mol cm^−3^), it can be seen that the slope of the curve (*I_p_/v*^0.5^) is positively correlated with the lithium-ion diffusion coefficient, and the larger the slope of the curve, the higher the diffusion coefficient of lithium ion [48]. As shown in Appendix A, Li–S cells assembled with G-CNT paper-based electrodes display a larger slope compared to other CNTs which implies the faster diffusion rates of lithium ions, manifesting the advantage of G-CNT for boosting polysulfides conversion kinetics. The fitted curve slopes of each CNT are demonstrated in Figure 4i; it can also be seen that OH-CNT and NH_2_-CNT also possess fast lithium-ion diffusion rates. Overall, the G-CNT electrode combines excellent electrical conductivity and a certain affinity for LiPS, which greatly contributes to the reaction kinetics of Li–S batteries. On the other hand, the OH-CNT and NH_2_-CNT electrodes have strong interactions with LiPS due to a large number of polar groups and also display great potential for high electrochemical performance.

Li–S batteries involve complex phase transitions during discharge, and the process of soluble polysulfide reduction to solid lithium sulfide contributes three-quarters of the theoretical capacity [13,49,50]. In addition, the deposition of the discharge product of lithium sulfide on the host electrode is based on non-homogeneous phase nucleation, and the study of the nucleation and growth of lithium sulfide on the electrode plays an equally important role in investigating the surface properties of the host material [51]. To understand the deposition process of Li_2_S on different surfaces, the nucleation currents were monitored. The current signal shows a peaking behavior as a function of time as shown in Figure 5a–e. The initial current drop can be attributed to the reduction of long-chain polysulfides to mid-chain polysulfides. After that, the current begins to increase since the nucleation of Li_2_S occurs, and the large succeeding current flow corresponds to the growth of the existing nuclei. Next, the current decays with time due to the overlap of the adjacent nuclei or their diffusion zones [37,41,52]. Comparing the nucleation curves of several CNTs, more Li_2_S was deposited on NH_2_-CNT (558 mAh g_s_^−1^), G-CNT (482 mAh g_s_^−1^), and OH-CNT (402 mAh g_s_^−1^); meanwhile, G-CNT and OH-CNT possess the highest peak current intensities during deposition, indicating that the -OH groups promote the nucleation and deposition of Li_2_S. The effect of -OH groups on the nucleation and deposition of lithium polysulfide is better.

To visualize the morphological evolution of the solid Li_2_S products, the electrode surfaces after the deposition and dissolution processes were performed by SEM. Figure 5f shows the electrodes after discharge at a constant potential. Obviously, the electrode surfaces with G-CNT and OH-CNT are mostly composed of isolated Li_2_S/Li_2_S_2_, indicating the discrete nucleation and anisotropic growth mechanism of Li_2_S/Li_2_S_2_ [37]. In contrast, the overall morphology of the solid product on the surface of MWCNT paper is smooth and compact, completely covered by Li_2_S/Li_2_S_2_, forming a dense passivation layer. Previous studies have demonstrated that lithium sulfide is more easily deposited on the surface of conductive polar materials, because they can provide an interface with lower interfacial energy for lithium sulfide, thus serving as a preferred non-homogeneous nucleation site [49]. CNTs, especially NH_2_-CNTs, OH-CNTs, and G-CNTs which are modified with polar groups on the surface, as the conductive fibers in paper electrodes, would have been highly susceptible to inducing the conversion of LiPS as well as Li_2_S nucleation. OH-CNTs and G-CNTs can form a uniformly distributed 3D fiber network structure with plant fibers in the preparation of electrodes because of the hydrophilic -OH effect. Moreover, the distribution of G-CNTs and OH-CNTs on the electrode surface is discrete, which means the isotropic growth of Li_2_S on G-CNT and OH-CNT electrodes allows for slow nuclei impingement and thus mitigated surface passivation [37]. In contrast, MWCNT is heavily distributed and agglomerated on the electrode surface, where the nucleation sites of Li_2_S on the electrode surface are continuous and easily tend to guide the formation of dense nuclei which easily merge and laterally grow to passivate the conductive surface [37]. While NH_2_-CNTs can both form fiber networks and be uniformly dispersed in pulp, some of them still aggregate on the electrode surface, so Li_2_S will thrive and passivate at the sites with CNT agglomeration, but precipitate and expose the conductive surface at the sites where well-dispersed CNT are isolated from each other.

Subsequently, the oxidation morphology of deposited Li_2_S was further investigated by using a normal galvanostatic charging process, as shown in Figure 5g. When the assembled cell is charged under a constant voltage of 2.35 V, the solid deposited layer is gradually dissolved and the conductive surface of CNTs appears again. Relatively fewer and smaller solid deposited particles remain on the surface of OH-CNT, indicating that the -OH groups not only have a better effect on the reduction reaction of sulfur but also are powerful in terms of catalyzing the oxidation of Li_2_S. Although G-CNT was also surface-modified with -OH groups, the small number of -OH groups resulted in a less catalytic effect. In conclusion, these results confirm that the deposition and dissolution of Li_2_S are significantly correlated with the distribution of CNTs and their surface functional groups.

The galvanostatic charge–discharge profiles of different batteries with a sulfur loading of 1.99 mg cm^−2^ (60 wt.% in cathode) at 0.25 C (1 mA cm^2^) are shown in Figure 6a, where the two discharge plateaus and one charge plateau match well with the two-step lithiation of sulfur and the corresponding reversible oxidation reaction. At 0.25 C, the G-CNT electrode shows a highest initial specific capacity of 1033 mAh g^−1^, which corresponds to a sulfur utilization of 61.6% and an area capacity of 2.05 mAh cm^−2^. The following is the OH-CNT electrode, which possesses an initial specific capacity of 965 mAh g^−1^, corresponding to a sulfur utilization of 57.6% and an area capacity of 1.91 mAh cm^−2^. Moreover, the OH-CNT and G-CNT batteries can be cycled for 200 cycles and still keep 81.3% and 77.4% of their initial capacity, respectively. In addition, as can be seen from the charge–discharge curves of the cells at the fifth cycle and the 200th cycle in Figure 6b,c, the discharge/charge plateaus of the G-CNT and OH-CNT cells are significantly longer and more stable compared with the other CNT-based cells, which shows a higher conversion rate of LiPS to solid Li_2_S/Li_2_S_2_ [53].

We also evaluated the rate capability of the paper-based electrode with CNTs at different current rates by means of galvanostatic charging and discharging. Figure 6d shows the cycle stability plots from these measurements, which confirms that the G-CNT paper overperforms the other CNTs at low current density. However, with an increase in current density, the specific capacities of all CNT cells start to decrease, and the specific capacity of the G-CNT battery even decreases to 30 mAh g^−1^ at 5 mA cm^−2^. It was suspected that the small amount of -OH groups and the porous structure of G-CNT may be not enough to support fast LiPS diffusion and conversion at high rates, resulting in a significant decrease in the battery capacity. In contrast, the OH-CNT cell displays a slow decrease in specific capacity as the current increases. It can be seen that ~800 mAh g^−1^ is still available at 5 mA cm^−2^, representing excellent rate capability. In addition, LiPS will not convert to mono-sulfur after cycles, but some of the insulating discharge product, lithium sulfide, will accumulate in the anode and even form a passivation layer. For the NH_2_-CNT and MWCNT electrode, in the sites of CNT agglomeration, the surface of the electrode may be more easily covered by a thicker and denser passivation layer composed of insulating discharge products after the charge–discharge cycles [51]. Therefore, the MWCNT and NH_2_-CNT batteries can only maintain a small portion of the capacity at high currents, which perhaps can be attributed to the less porous, non-polar structure of MWCNT, the non-uniform distribution of CNTs, and more severe passivation of their paper-based electrodes. It is worth noting that, as demonstrated in previous sections, -OH groups can promote the conversion of long-chain LiPS and further reduce it to short-chain insoluble lithium sulfide [46], and this process corresponds to the low-discharge plateau curves of Li–S batteries. Therefore, it can be observed in the charge–discharge curves of Figure 6e–i that the curves of OH-CNT paper-based batteries can still maintain a relatively complete discharge profile at high current densities, whereas most of the discharge processes of other CNTs show lower capacities because of the significant polarization.

## 4. Conclusions

In summary, the influence of integrating carbon nanotubes into paper-based sulfur cathodes significantly impacts both its mechanical and electrochemical properties. In this study, we fabricated paper-based sulfur cathodes by incorporating diverse CNTs and plant fibers through a papermaking process. Specifically, CNTs with surface-modified functional groups and optimal aspect ratios were chosen as the conductive fibers. These fibers form a 3D network structure, providing a highly conductive and structurally stable framework for electrochemical reactions of Li–S batteries.

Among these different CNT-based electrodes, G-CNT paper shows the best electrochemical performance under low current due to its good conductivity and small number of -OH groups, delivering the highest initial specific capacity of 1033 mAh g^−1^ at a current of 0.25 C with a sulfur loading of 1.99 mg cm^−2^. However, G-CNTs are not suitable for high-current charge–discharge conditions because of the lean -OH groups that are not sufficient to capture LiPS and facilitate its fast conversion. NH_2_-CNTs have polar functional groups of -NH_2_ that exhibit high affinity for LiPS, but they are unevenly distributed on the electrode during papermaking, leading to severe surface passivation in charge–discharge cycles, and thus affecting the battery performance. OH-CNTs are fully dispersed during the preparation of paper-based electrodes due to the large amount of -OH groups on the surface, forming a uniform, porous and stable 3D network structure with plant fibers, which not only possesses superior mechanical properties but also help to capture LiPS and facilitate its fast conversion. Thus, even after cycles under high currents, the cycling stability and rate capacities of OH-CNT electrode outperform other different CNT-based electrodes. Moreover, the Li–S cells containing OH-CNTs show the highest capacity of 802 mAh g^−1^ at 5 mA cm^−2^ with 1.99 mg cm^−2^ sulfur loading, which indicates much greater potential for the high performance of Li–S batteries in the future.

## Figures and Tables

**Figure 1 nanomaterials-14-00484-f001:**
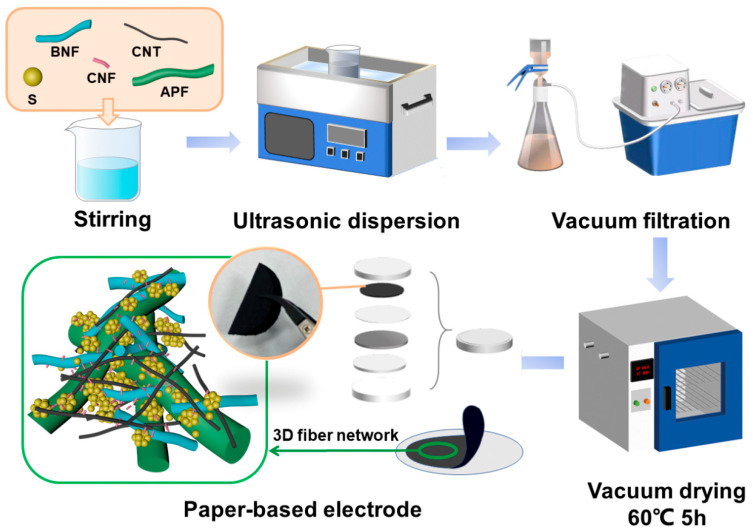
Schematic representation of the preparation process of a paper-based sulfur cathode and its internal 3D network structure.

**Figure 2 nanomaterials-14-00484-f002:**
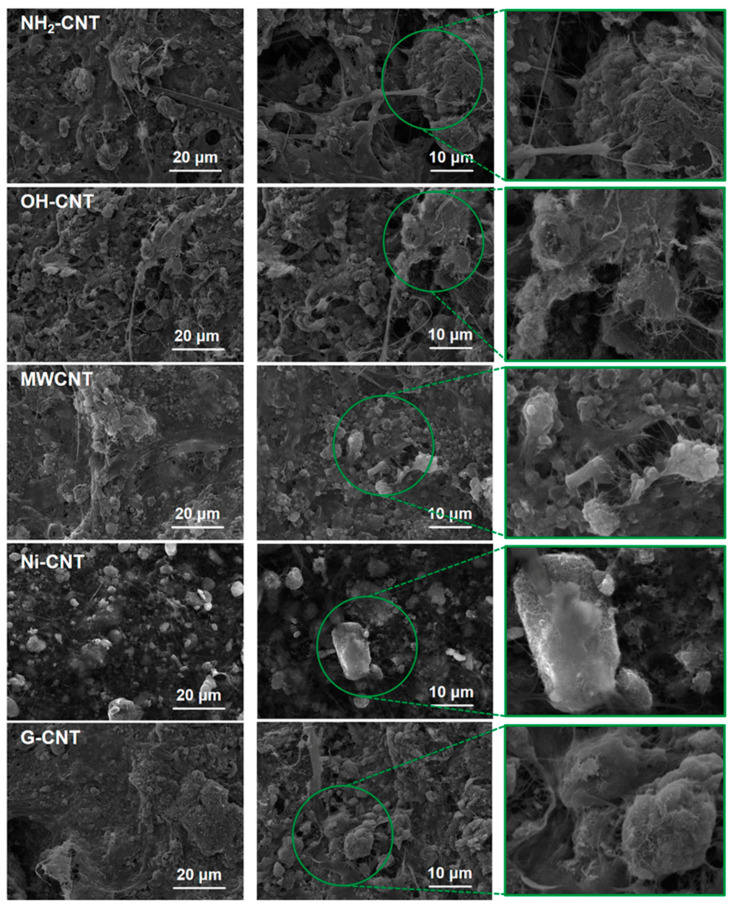
SEM images of paper-based sulfur cathodes with different CNTs.

**Figure 3 nanomaterials-14-00484-f003:**
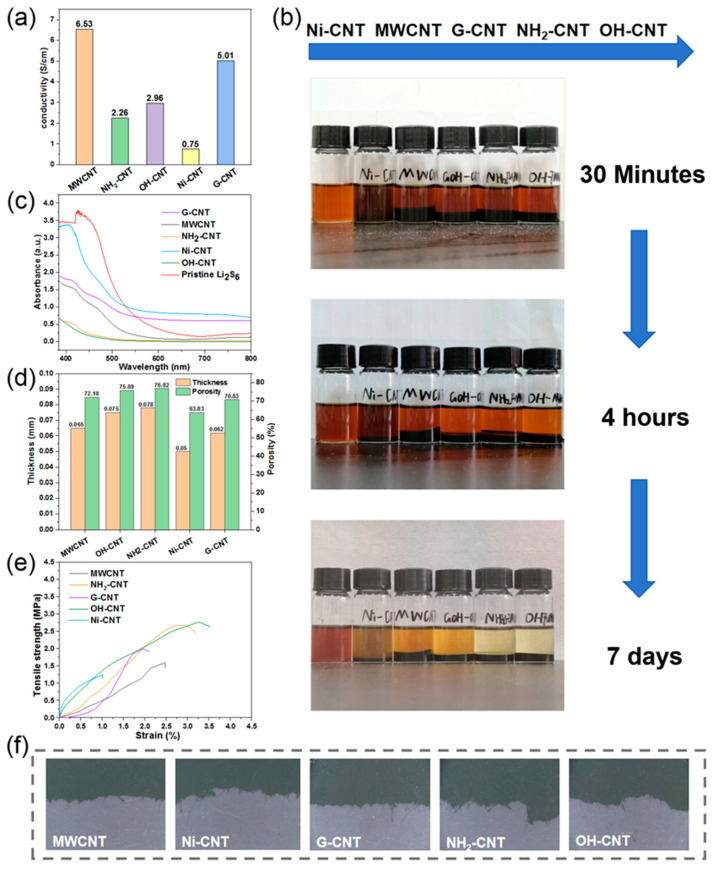
(**a**) Conductivities of paper-based electrodes with different CNTs. (**b**) Optical images of CNTS soaked in the 5 mM Li_2_S_6_ solution. (**c**) UV-vis absorption spectra of different CNTs in 5 mM Li_2_S_6_ solution after 7 days absorption. (**d**) Thickness and porosity of different electrodes. (**e**) Stress–strain curves of paper-based electrodes. (**f**) The images of the electrode fracture site after stretching.

**Figure 4 nanomaterials-14-00484-f004:**
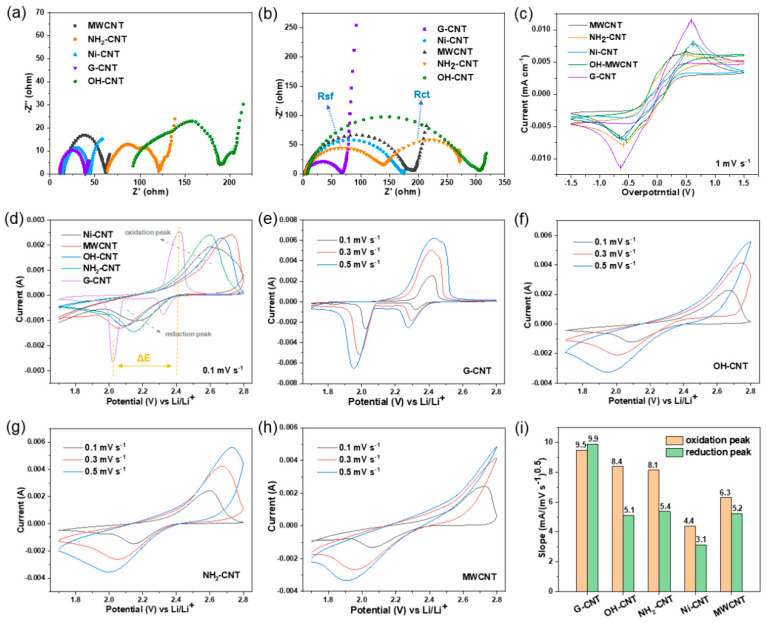
EIS patterns of (**a**) symmetric cells and (**b**) Li–S cells with different CNT papers. (**c**) CV curves of the symmetric cells at a scan rate of 1 mV s^−1^ with different CNT papers. (**d**) CV curves of the Li–S cells at a scan rate of 0.1 mV s^−1^. CV curves of Li–S cells assembled by (**e**) G-CNT paper-based electrode, (**f**) OH-CNT paper-based electrode, (**g**) NH_2_-CNT paper-based electrode, and (**h**) MWCNT paper-based electrode at scan rates of 0.1 to 0.5 mV s^−1^. (**i**) The fitted slopes of the CV peak current versus the square root of the scan rate.

**Figure 5 nanomaterials-14-00484-f005:**
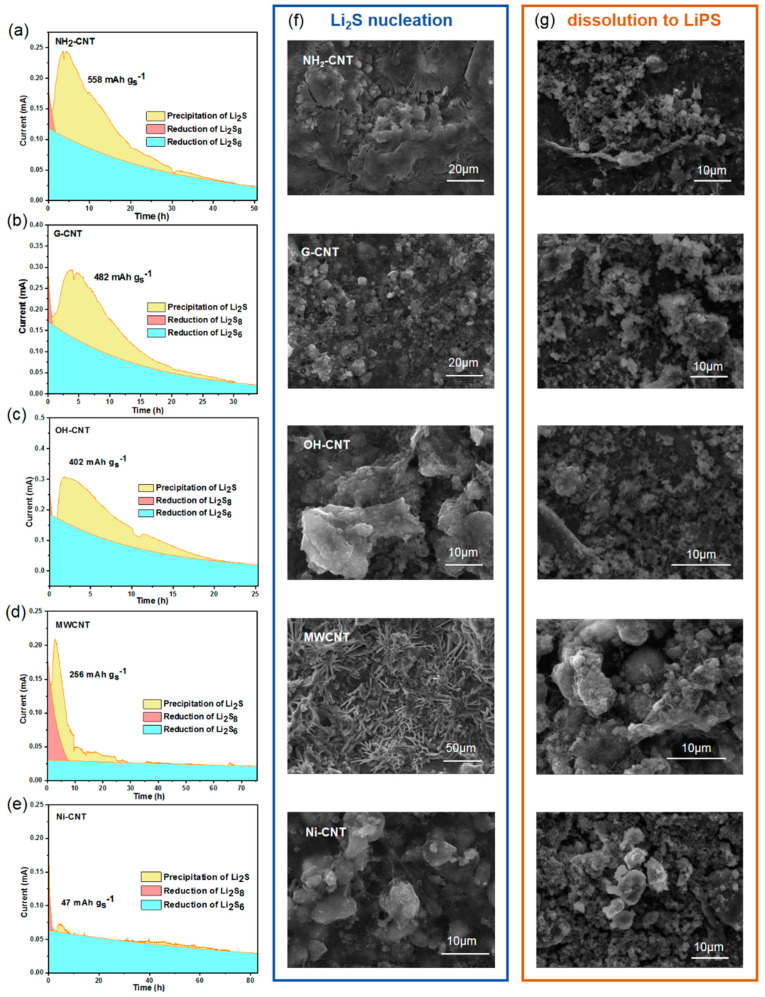
Li_2_S nucleation curves of (**a**) NH_2_-CNT paper, (**b**) G-CNT paper, (**c**) OH-CNT paper, (**d**) MWCNT paper, and (**e**) Ni-CNT paper. Corresponding electrode morphologies (**f**) after Li_2_S nucleation and (**g**) dissolution to LiPS on the surfaces of electrodes.

**Figure 6 nanomaterials-14-00484-f006:**
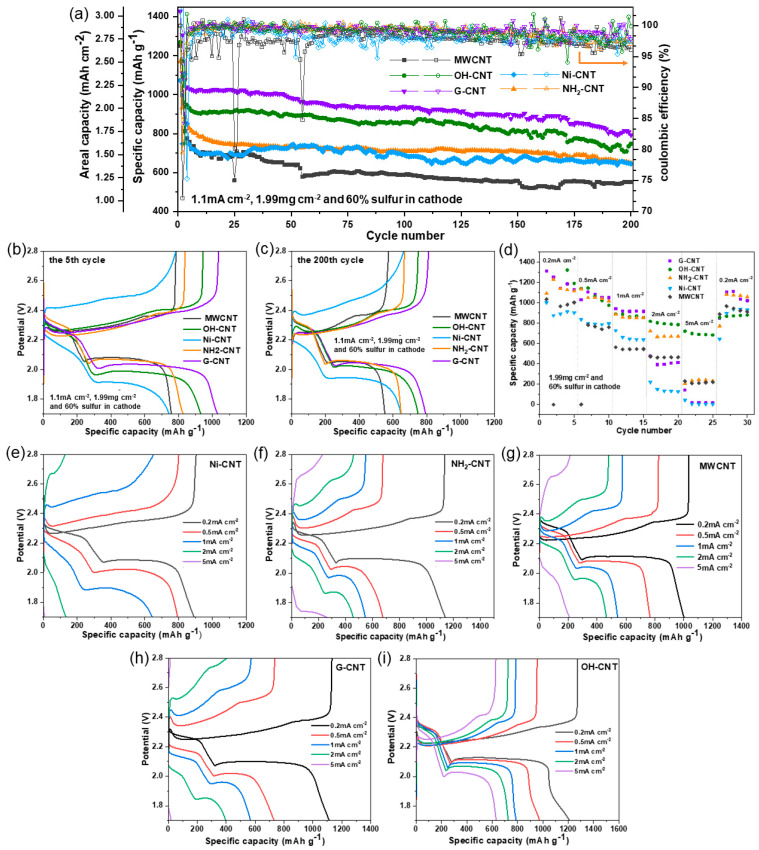
(**a**) Cycling performance of different CNT paper-supported Li–S cells with sulfur loading of 1.99 mg cm^−2^ at 0.25 C. (**b**) The fifth cycle curves and (**c**) the corresponding 200th cycle curves at 0.25 C of different CNT paper-supported Li–S cells with sulfur loading of 1.99 mg cm^−2^ (**d**) Rate performance of different CNT paper-supported Li–S cells with sulfur loading of 1.99 mg cm^−2^. Discharge-charge curves of (**e**) Ni-CNT, (**f**) NH_2_-CNT, (**g**) MWCNT, (**h**) G-CNT, and (**i**) OH-CNT paper-supported Li–S cells with sulfur loading of 1.99 mg cm^−2^.

## Data Availability

Data are contained within the article and Appendix A.

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
