# Peer review of "Investigating the Influence of Diverse Functionalized Carbon Nanotubes as Conductive Fibers on Paper-Based Sulfur Cathodes in Lithium–Sulfur Batteries"

_nanomaterials, 2024, doi:10.3390/nano14060484_

Round 1

Reviewer 1 Report

Comments and Suggestions for Authors

General: This is a nice piece of work in which low-cost, scalable paper-making methods are applied

to the fabrication of Li-S battery materials.  The use of two different kinds of cellulosic fiber is

a clever move.  I do have some questions and comments; see below.

P9:  What does this statement,

After these tests, we found that the flexible sulfur electrodes prepared with different

functionalized CNTs have their characteristics in terms of conductivity, LiPS adsorption,

porosity, and mechanical properties.

mean?

Fig. 4 and others: I find the arrangement a bit confusing.  The graphs are arranged something like this:

* + *

+ - -

- - -

x x y

* = EIS plots

+ = voltammograms, all samples at fixed rate

- = voltammograms, single sample, variable rate

x = Randles-Sevcik plots

y = fitted slopes of Randles-Sevcik plots

I suggest that only some of these curves be presented in the main paper and the rest in SI, which

seems to be allowed by this journal.  For instance, the Randles-Sevcik plots could go in SI, with the

bar graph of coefficients in main text.  By comparison, Fig. 5 is nicely laid out, though the blue and

brown frames could be eliminated.

P10:  In the Randles-Sevcik equation, it's stated that n is the 'number of electrons'.  Number

of electrons per what?  What are the units?  Is the numerical coefficient 2.65e5 dimensionless or

does it have units?

English:  It's a bit off.  For instance, on P13, a sentence ends,

the solid deposited layer is gradually dissolved and the conductivity surface reappears.

I suspect that 'conductivity' should be 'conductive'.

Figure 6a: What's the difference between the open and closed symbols?  I don't see that mentioned

in the caption or text.

Comments on the Quality of English Language

See suggestions for authors.

Reviewer 2 Report

Comments and Suggestions for Authors

I have reviewed the original research article entitled “Investigating the Influence of Diverse Functionalized Carbon Nanotubes as Conductive Fibers on Paper-Based Sulfur Cathodes in Lithium-Sulfur Batteries” work by the authors C. Xiong et al., submitted to Nanomaterials / MDPI.  The stated lithium-sulfur battery is one of the most promising candidates for becoming the battery system of the near future. Its theoretical gravimetric energy density of 2500 Wh kg-1 is over four times higher than the ideal state of common lithium-ion batteries.  Also, the Li- S system is getting wide attention due to its high energy density, natural abundance of sulfur, and low environmental impact. Li–S redox involves multi-step chemical and phase transformations between solid sulfur, liquid polysulfides, and solid lithium sulfide (Li2S), which give rise to unique challenges in Li–S batteries. Therefore, the area of research in the proposed work is significant. The unavoidable shuttle effect and the inability of traditional components to withstand (bending) have brought challenges to this area of research. The authors have attempted to address this through various conductive fibers on paper-based Sulfur cathodes in the proposed LI-S system which resolves the Li2S agglomeration.

The following points must be addressed:

·         The abstract is like storytelling with no performance metrics reported from the work that the authors carried out.

·         Abstract, line 7: Use either “Herein” or “in this study”

·         True, traditional LIBs have limitations in the theoretical energy density but arguably recently reported Al-air batteries 10.1039/D3DT03736C have exceeded specific capacity. Include this in the discussion.

·         The reported value in the introduction (page 2, line 3) is not specific capacity but energy density. However, please also include the specific capacity values.

·         What are flexible sulfur cathodes?

·         Page 2, paragraph 3, second sentence “This 3D fiber network…it is too long sentence.

·         The novelty of the current work must be emphasized in the last paragraph of the introduction.

·         Well, SDBS is a surfactant, elaborate on the acronym.

·         What is the rationale for choosing LITFSI electrolyte?

·         Page 8, “the small amount of -OH groups on the surface of G-CNT are not enough to form a large number of hydrogen bonds” this could be referred to the literature 10.1039/D0DT01871F.

·         The shape of the CV curves and the redox peak shifts observed in Figure 4 must be addressed.

·         From Fig 4a, report the Rs and Rct values.

·         At this time, Rini is much larger than Rct – what is Rini and include their values.

·         What does the peak separation imply?

·         What is the best charging rate for the constructed Li-S cell?

·         The last line in the discussion “end with the end of the high discharge plateau” is unclear.

·         In the section conclusion, include the specific capacity values and cell potentials obtained for the different electrodes.

Round 2

Reviewer 2 Report

Comments and Suggestions for Authors

The revised version is suitable for publication.